# Tensile Strength and Structure of the Interface between a Room-Curing Epoxy Resin and Thermoplastic Films for the Purpose of Sensor Integration

**DOI:** 10.3390/polym13030330

**Published:** 2021-01-21

**Authors:** Alexander Kyriazis, Riem Kilian, Michael Sinapius, Korbinian Rager, Andreas Dietzel

**Affiliations:** 1Institut für Mechanik und Adaptronik, Technische Universität Braunschweig, 38106 Braunschweig, Germany; r.kilian@tu-braunschweig.de (R.K.); m.sinapius@tu-braunschweig.de (M.S.); 2Institut für Mikrotechnik, Technische Universität Braunschweig, 38124 Braunschweig, Germany; k.rager@tu-braunschweig.de (K.R.); a.dietzel@tu-braunschweig.de (A.D.)

**Keywords:** interface, interphase, adhesion, sensor integration, fibre reinforced polymer, epoxy, thermoplast

## Abstract

The article presents a study on the adhesion of thermoplastic films to a room temperature-hardening epoxy resin, which deals with an important question on sensor integration into fibre composites. By means of a morphological box, a test specimen is developed, which allows to test strength values for the adhesion of thermoplastic films to epoxy resin. Polyimide (PI), which is typically used as a carrier material for flexible sensors, is compared with the thermoplastics polyetherimide (PEI), polyethersulfone (PES) and polyamide 6 (PA6). To evaluate the spatial formation of the interface, images taken with a light microscope, fluorescence microscope and electron microscope and an energy-dispersive X-ray spectroscopy (EDX) analysis are presented. The images show that during the curing process of the epoxy resin the initially expected pronounced interphase does not form. In this respect, it is surprising that PEI achieves such a high adhesion strength even without extended interphase formation, that the failure of the test specimen occurs in the epoxy resin region at a tensile stress of 70 MPa and not at the interface between epoxy and PEI, as might initially be assumed. It is also surprising that PES exhibits the lowest adhesion strength of 5 MPa to room temperature-hardening epoxy resin, although in previous investigations it was often used as a soluble toughness modifier for epoxy resins. The tensile adhesion strength of PI to epoxy resin was found at 27 MPa and the tensile adhesion strength of PA6 to epoxy resin was found at 13 MPa. For sensor integration, the findings mean that flexible sensors on PEI substrates promise a low tendency to delaminate even in the room temperature-hardening epoxy resin used, while the other materials tested indicate an increased tendency to delaminate.

## 1. Introduction

The combination of thermoplastic films with an epoxy resin matrix can be useful for a wide range of applications. On the one hand, the thermoplastic film can be applied for toughness modification to increase the crack propagation resistance of fiber-composite components [1]. On the other hand, thermoplastic films have been investigated as an electrically insulating layer for the production of multifunctional capacitors to store capacitive energy in composite components [2]. Schuett et al. report on an application in which films of polyetherimide (PEI) and polyethersulfone (PES) were incorporated into composites to improve fire resistance [3]. In his dissertation, Kaps investigated thermoplastic films for combining prepreg processes with resin infusion processes [4]. Bruckbauer et al. proposed thermoplastic films in general as carriers of functional layers [5,6].

The findings presented in this article originate from a research project in which film sensors based on a suitable thermoplastic material film are integrated into a fiber-composite. In the sensor integration, there can be different measured variables, the present research pursues the goal of integrating the sensors into the fiber-composite for cure monitoring. The monitoring of the entire curing process enables a precise determination of the degree of crosslinking, allowing to obtain the best achievable mechanical properties of the epoxy resin [7,8,9]. This goal is countered by the tendency of integrated sensors to behave as a foreign body [10]. PI (Kapton) is a very typical substrate material for flexible sensors [11] but shows low adhesion to the surrounding composite. Figure 1 highlights the mechanical impact of film sensors as an important issue for sensor integration and also shows different other questions arising from the integration of film sensors for cure monitoring purposes. The mechanical impact has already been observed in earlier experiments on critical energy release rates [12]. One way to reduce the weakening effect is to provide holes in the film through which the epoxy resin can crosslink [7,13,14]. Another approach is to produce films out of plasticized epoxy resin [15]. Instead, however, thermoplastics can also be used as substrate materials, which show stronger adhesion to the epoxy resin and ideally even dissolve in the epoxy resin, so that the foreign body introduced completely disappears during production. Therefore the interphase formation is also highlighted in Figure 1 as a question to be discussed in the present article.

There is no unified model for predicting the quality of adhesion, but rather a variety of theories, each of which takes different mechanisms into account [16]. In the case of polyamide 6 (PA6) there is good compatibility between several molecular functional groups of the thermoplastic and the epoxy resin [17]. Previous investigations have already shown that integrated PA6 films have little effect on the interlaminar shear strength and can significantly increase the critical energy release rate in Mode I [12]. Compounds between polymers such as epoxy resins and thermoplastics can enable a special form of adhesion, namely a so-called interphase developed between both materials. In this process, both materials diffuse into the other material within a certain spatial range so that the thermoplastic film dissolves at the contact surface, resulting in a gradient transition between the two materials. The condition for diffusion is that the thermoplastic used is soluble in the epoxy resin. Such an interphase formation has already been reported in the literature for the thermoplastics PEI and PES [5,18,19,20,21]. In our own preliminary tests on these thermoplastics embedded in prepreg specimens, no impairment of the interlaminar shear strength and a significant improvement of the critical energy release rate in Mode I was observed for these thermoplastics, with PEI being significantly better than PES [12].

The literature reports the following mechanism for interphase formation. At the beginning of the interphase formation, the monomers of the epoxy resin diffuse into the thermoplastic, forming a thermoplastic region that is in a gel-like state [18]. Individual chains are released from the gel, which diffuses into the epoxy resin component to form a so-called liquid layer [19]. In this way, a concentration gradient of the thermoplastic component is created. In the case of PES, this concentration gradient persists and both components remain mixed together during further curing of the epoxy resin [20]. In the case of PEI, a reaction-induced decomposition occurs, which leads to a spatially varying interphase [18,20]. Due to the spatially varying concentrations, locally different segregation mechanisms like binodal and spinodal decomposition come into play. Depending on the segregation mechanism, the temperature history and the thermoplastic content, different morphologies will occur [22]. For this reason, an interphase is reported for PEI, starting from epoxy resin and characterized first by PEI islands in epoxy, then a bicontinuous structure, epoxy islands in PEI and finally pure PEI [18,21]. Pearson et al. attribute increases in the energy release rate to various mechanisms such as crack deflection [23]. The conclusion of Bruckbauer et al. is different: They attributed the increase in toughness to the mere presence of the thermoplastic as a tough material. The exact shape of the interphase is not important for the toughness increasing effect according to Bruckbauer et al. [5].

While the fracture toughness of the bond between epoxy resin and thermoplastic is a parameter frequently investigated in the literature, there are few studies dealing with the strength of the interface between flexible carrier films for integrated sensors and epoxy resin. The few articles dealing with the determination of strength values almost exclusively use interlaminar shear specimens. This article presents tensile strength values for the interfaces between different thermoplastics and epoxy resin and investigates the relation to the interface formation.

## 2. Materials and Methods

The epoxy resin used consists of 100 weight parts of the resin component RIMR 426 together with 26 weight parts of the hardener component RIMH 435. Both components come from Hexion (Columbus, OH, USA). The resin is a mixture of the diglycidyl ether of bisphenol A, and the diglycidyl ether of bisphenol F and 1,4-bis(2,3-epoxypropoxy)butane. The hardener component is a mixture of the aminic hardeners 3-aminomethyl-3,5,5-trimethylcyclohexylamine, trimethylhexane-1,6-diamine, polyoxypropylenediamine and m-phenylenebis(methylamine). The epoxy resin is suitable for curing at room temperature and was processed at room temperature in the tests.

The following thermoplastic films were used: The PA6 film used is a 25 μm thin cast polyamide film which was kindly provided by mf-folien GmbH (Kempten, Germany). As PEI a 25 μm thin Ultem 1000 film from Goodfellow (Huntingdon, United Kingdom) was used. The PES film used is the 25 μm thin Lite S film from Lipp Terler (Gaflenz, Austria). The PI film is a 50 μm thin Kapton film type HN from Dupont (Wilmington, DE, USA).

A test specimen was developed to examine the strength of the bond between the thermoplastic films and the epoxy resin. Therefore different concepts were elaborated in a morphological box. In the preliminary phase of the concept design, only the film materials and the epoxy resin system had been determined as well as the requirement to produce the test specimens from liquid resin by a casting process without fiber material. The sub-functions for the mechanical testing-method are divided into the stress type occurring in the interphase and the load type introduced into the specimen by the testing machine. Yet, there are no test standards for the analysis of the mechanical properties of this type of polymer combination. Therefore, testing standards for adhesion in composites, in particular adhesive bonding of plastics, have been considered. In this case, an adaption is possible due to acting adhesive forces between the epoxy-rich and thermoplastic-rich phase. The specimen geometry is characterized by the type of joint which describes the structural arrangement of the contact zone between the thermoplastic film and the epoxy. Regarding the specimen manufacturing the casting mold and its number of components, the casting process and the arrangement of the thermoplastic film are considered.

Subsequently, the part solutions for the selected part-functions are sought and presented in the morphological box, see Scheme 1. Three concepts have been combined from the morphological box and the concept selected is highlighted in Scheme 1. The concept provides a specimen in which tensile stress builds up in the interface due to a pulling force of the testing machine. The interface between the thermoplastic film and the epoxy resin has the shape of a butt joint. The resin components are mixed according to the manufacturer’s instructions as mentioned above and then poured in one single casting step into an open 3D-printed casting mold which is divided into two similar halves. The thermoplastic film is clamped vertically between the two halves of the mold. Specimens without integrated thermoplastic film consisting of pure resin serve as a reference in the experiments. The resin is cured for 24 h at room temperature. Later the specimens are cut with a band saw into 4 mm thick and 10 mm wide rods. The adapted specimen with the integrated thermoplastic film is shown in Figure 2. For the specimens in which bad adhesion was expected the rods were sanded to provide a smooth surface, see Figure 2a. The specimens in which strong adhesion was expected were also narrowed in a sanding process to avoid breakage in the clamping region of the specimens during the experiments to a minimum width of 6 mm, see Figure 2b. The non-tapered prismatic specimen geometry is more desirable than the tapered geometry because of the more uniform stress state and because the contact zone, which is weak in some specimens already, is not weakened further. Therefore, only those specimens where otherwise a fracture in the clamping would have been expected were tapered.

For the testing of the specimens an Instron (Norwood, MA, USA) 5567 universal testing machine with a load cell up to 10 kN is used. The specimens were pulled with a traverse speed of 0.5 mm/s. The invented specimen envisages the evaluation of the tensile strength of the interphase according to DIN EN 15870. The tensile strength of the interphase is then given by:(1)$$\sigma_{y}=\frac{F_{B}}{t\;\cdot \;b}$$
where $\sigma_{y}$ stands for the tensile strength, $F_{B}$ for the breaking load, *t* for the specimen height and *b* for the specimen width.

In addition, various imaging techniques were used to investigate the interphase. For this purpose, grinding patterns were created by first embedding cross-sections in acrylic resin, then grinding and polishing them in different polishing steps with the finest grain size of 0.25 μm. The polished specimens were first examined under a Keyence (Osaka, Japan) digital light microscope VHX-7000 and with the aid of a Leica (Wetzlar, Germany) M205FA fluorescence microscope in various wavelength ranges. In addition, the samples were examined under a Zeiss (Oberkochen, Germany) Evo scanning electron microscope (SEM) and the spatial distribution of various elements was studied using energy-dispersive X-ray spectroscopy (EDX). The fluorescence microscope was used because of the higher achievable contrast between thermoplastic and epoxy regions and the electron microscope was used because of the high resolution. The basic idea of fluorescence microscopy is to excite the sample in a narrow wavelength range and to measure the light emission due to fluorescence in a slightly longer wavelength range. The fluorescence of the material leads to a wavelength shift towards higher wavelengths and lower frequencies and energies. Since it occurs material-specifically in certain excitation frequency ranges, it is usually possible to find a wavelength range for a given material pairing in which one component fluoresces and the other component barely fluoresces at all. In this way, fluorescence microscopy achieves high contrasts.

For fluorescence microscopy, the excitation wavelength ranges between 375 and 435 nm (violet), 490 and 510 nm (blue) were used, 460 and 500 nm (blue-green), 541 and 551 nm (green), 542 and 582 nm (yellow-green), and between 590 and 650 nm (red). The reflected light signals are filtered so that only light in the response wavelength ranges between 450 and 490 nm (blue to green), 520 and 550 nm (green to yellow) is used, 512 and 542 nm (green), 565 and 605 nm (yellow), 604 and 644 nm (orange) and between 662 and 738 nm (dark red) are transmitted and recorded. The different wavelength ranges are summarised in Table 1. A resolution of about 0.3 μm per pixel is achieved at a 20× magnification of the microscope. By trying out the different wavelength ranges, different areas of high contrast were found for different thermoplastics.

In order to examine the specimens under the SEM subsequently, the surface of the ground specimens for microscopic investigation is sputtered with gold for 180 s so that it is not charged by the electron beam. The accelerator voltage of the SEM is 15 kV. The electron microscope allows on the one hand the scanning of the surface with very high magnifications and on the other hand the spatially resolved analysis of the characteristic X-ray spectrum, known as EDX analysis. From the X-ray spectra, the spatial distribution of different atoms can be concluded. In EDX analysis, it should be noted that the electron beam impinging on the sample excites not only one point on the sample, but a bulb-shaped area with a diameter that depends on the accelerator voltage and the atomic weight of the atoms under observation and is approximately in the lower single-digit micrometer range [20]. Due to this resolution limitation, measurements were made using 100 measuring points over a measuring length of 100 μm, resulting in measuring point distances of about 1 μm.

## 3. Results

Figure 3 shows the strength of the interface between the epoxy resin and thermoplastic measured in the tensile test. The error bars mark the 95% confidence intervals from the one-sample *t*-test. At least five specimens were tested for each configuration. The data show that in the case of PES, PA6 and PI the interface between thermoplastic and epoxy resin does not reach the tensile strength of the reference samples. The significantly weakest bond is the interphase between the PES film and the epoxy resin. In the case of PEI, the interphase reaches the strength of the reference samples and in the bar chart, it can be seen that the measured strength values of the reference samples and the PEI-epoxy interface do not differ significantly. As the fractures occurred in the surrounding epoxy resin in four of five specimens with integrated PEI-films, the measured strength value only represents a lower estimate of the real strength of the interface.

The comparison of typical tensile curves of different test specimens shows an arc in the stress-traverse path curve of the reference specimens before failure, see Figure 4. For the test specimens made of PES, PA6 and PI such an arc cannot be observed and the failure occurs at much lower stresses due to a sudden collapse of the mechanical stress. The test specimens with integrated PEI film, on the other hand, exhibit the same arc as the reference test specimens, which can also be explained by the fact that it is not the interface that fails, but the neighboring epoxy resin. It is therefore obvious that similar behavior can be observed in the stress traverse path characteristic curve. For this reason, the measured strength values only represent a lower estimate of the true bond strength. Differences in the rates of increase in the diagram are caused by the fact that the non-tapered specimens have different stiffnesses compared to the tapered specimens. A slight variance within the group of the tapered specimens or the non-tapered specimens is caused by the variance in clamping in the tensile testing machine and by variance in the grinding process of the specimens.

The subsequent microscopic investigation was intended to find a relation between interphase formation and the strength value obtained in the tensile test. Figure 5 shows light microscope images of the films embedded in epoxy resin. All images in this article are aligned in such a way that the thermoplastic film is visible as a vertical bar in the center of the image. The illustrations show a sharp transition from the thermoplastic film material to the surrounding epoxy resin for all film materials. Extended interphase is not visible on the light microscope images. Within the tolerances, the films have the same thickness as before embedding, 50 μm for the PI film and 25 μm for all other thermoplastic films used. It is only noticeable that during the grinding of the micrograph samples with integrated PA6 film, a grinding burr developed on the right edge of the film, see Figure 5a. This burr is probably due to the high plastic elongation of the film materials compared to other thermoplastic materials. In addition, larger craters are visible in the texture of the surrounding epoxy resin in the test specimens with PA6 film than in the other grinding specimens. An explanation for this could also be an interaction between the brittle epoxy resin and the PA6 films with very high ductility.

While the contrast in light microscope images is created by the optical transmission of thermoplastic materials, fluorescence microscopy uses the physical property of fluorescence in different frequency ranges. Due to their solubility in epoxy resin, an interphase was initially expected for the thermoplastics PEI and PES and the sharp limit recognizable in the light microscope images (Figure 5) was doubted. In contrast, no interphase was expected for PI and PA6, as the literature reports no solubility. For this reason, fluorescence microscope images of the interfaces between PEI or PES and epoxy resin are compared with fluorescence microscope images of the interface between PI and epoxy resin in Figure 6. Fluorescence microscope images of the samples with PA6-foil were not taken, because firstly no interphase was expected in these samples and secondly a sufficient contrast could not be achieved in any of the existing wavelength ranges. In other words, a fluorescence microscope for the investigation of the PA6 epoxy interface does not offer any advantages over a light microscope. Since the fluorescence microscope produces contrast with a different physical effect than a light microscope, it should be excluded by taking fluorescence microscope images that an interphase in Figure 5 might have been overseen due to lack of contrast. Particularly good contrasts were obtained for the interface between PEI and PI and epoxy resin with the setting TXR with excitation in the wavelength range from 542 to 582 nm and emission in the wavelength range from 604 to 644 nm and for the interface between PES and epoxy resin with the setting L5 with excitation in the wavelength range from 460 to 500 nm and emission in the wavelength range from 512 to 542 nm.

The fluorescence microscope images of the different thermoplastic films are shown in Figure 6. As the coloring of the images obviously does not match the given wavelength ranges, it should be noted that the camera takes monochromatic images and displays them in a given color. Because the color green was chosen for the display, all images appear in green, even if some of the images were taken in the red spectral range. The images show that in all three cases there is a sharp border between the thermoplastic film and the surrounding epoxy resin. A thickness measurement of the films confirms the findings from the images taken with the light microscope, so that no resolution of the films can have taken place. If an interphase formation had taken place, an island-sea morphology with an extension in the two to three digit micrometer range would be expected, as already reported in the literature: for example 80 μm [18], 113.5 to 188.7 μm [5]. Thus, fluorescence microscopy confirms the assumption, which is also suggested by light microscope images: No distinct interphase was formed in the tensile test specimens.

In order to be able to resolve even very small structures, images were taken with an electron microscope in addition to those taken with a light microscope and the fluorescence microscope. The electron microscope also offers the option of examining the interphase with energy dispersive X-ray (EDX) analysis. Figure 7 shows the images produced with a scanning electron microscope at a magnification of 1000. The images have a higher resolution than the microscope images, but do not show any further details that could not also be seen on the light microscope images or the fluorescence microscope images. Figure 7a shows again the grinding burr and the more fissured epoxy resin texture compared to the other specimens. For all the film materials used, a sharp border between the epoxy resin and the thermoplastic film can be seen. In the grinding specimens with PI-film and with PES-film, however, it can be seen that a crack has developed between the film and the epoxy resin, possibly due to forces exerted on the specimens when the embedding resin shrunk or during grinding. These cracks again underline the very low strength values of the bond to the epoxy resin, which were determined in the tensile tests. They are not visible on the images in Figure 5 and Figure 6 because the images from SEM and the images from light microscopy or fluorescence microscopy do not show the exact same locations of the boundary zone. Examination with the electron microscope also shows no signs of extensive interphase formation.

Figure 8 shows the result of the spatial EDX analysis carried out along the red arrow in the upper part of the figure. Along the 100 μm long arrow 100 measuring points were arranged. The results show that the sample contains carbon, nitrogen, oxygen, sulfur and small amounts of chlorine. As the specimen was sputtered with gold before SEM examination, gold is also detectable by EDX; however, because gold presence could clearly be traced back to the sputtering, the element gold was excluded from the results. In the area of the foil, a slightly reduced carbon content and a significantly increased sulfur content can be seen. On closer inspection, it can also be seen that the chlorine content in the area of the film (between 35 and 62 μm), which is already very low in the epoxy resin, is reduced to zero. The fact that the carbon and sulfur content changes in the area of the transition at about 62 μm in a range of several micrometers is not a sure sign of interphase, but can also be attributed to the mediating effect of the EDX, which was already mentioned in the Materials and Methods section. Vandi showed in a Monte Carlo simulation that the excitation bulb reaches a width of about one micrometer at an accelerator voltage of 10 kV [20]. The width of this excitation bulb continues to increase as the accelerator voltage increases and in the experiment, an accelerator voltage of 15 kV was used. Therefore, the smooth transitions at the boundary of the film are more likely to be due to process-inherent averaging than to an interphase, which would not have been visible in all microscope images shown so far. The strong fluctuations of all signals in the range between 20 and 35 μm which can be recognized in Figure 8 are due to the gap in the upper part of the figure. Therefore the data between 20 and 35 μm cannot be interpreted.

The chemical composition of the epoxy resin suggests no sulfur content in the epoxy resin. The origin of the low constant sulfur content far away from the integrated thermoplastic film has not yet been definitely clarified. In the EDX investigations of the other thermoplastic films in Figure 9, the evaluation of the sulfur signal was omitted because here no sulfur was expected to occur. Figure 9a shows, due to the grinding burr (see Figure 7a) in the range around 40 μm, strongly oscillating curves from which no information about the border zone between the PA6 film and the surrounding epoxy resin can be deduced. In the range between 45 and 67 μm to the right, it is noticeable that the nitrogen content is higher than the oxygen content. The PA6 film lies in this range and the slightly increased nitrogen content is plausible for PA6. In Figure 9b a relatively constant course can be seen for carbon, nitrogen and oxygen. On closer inspection, it can be seen that the determined chlorine content is slightly above zero in the range from 0 to 22 μm and decreases to zero in the range from 22 to 72 μm. This corresponds to the range in which the PI film is located. At about 72 μm, strong fluctuations in all signals are observed, which are due to a gap between the epoxy resin and the PI film, as already discussed for the test specimens with PES film. In Figure 9c no abnormality can be seen. The film cannot be identified from the contents of carbon, nitrogen and oxygen. Figure 8 therefore shows primarily that the EDX analysis is useful for investigating interphases between PES films and epoxy resin due to the differences in the sulfur content. However, looking at Figure 9, it becomes clear that the method is not suitable for film materials with a composition similar to that of the epoxy resin.

## 4. Discussion

The three microscopic methods used to investigate the interface between the epoxy resin and the used thermoplastic films do not show any evidence of extensive interphase formation in the two to three-digit micrometer range, as has been reported so far [5,18]. The transitions of atomic fractions observed with EDX in the interface area between thermoplastic film and epoxy resin can also be attributed to the process-inherent mediating effect of EDX analysis. In any case, the transitions cannot prove interphase formation. To prove or disprove an interphase formation in the submicrometer range, not all applied methods are suitable without restrictions. The light and fluorescence microscopes used are limited by their resolution, and the scanning electron microscope offers a low contrast between the different materials, so that a higher resolution is possible but not the differentiation of the materials. Here, an etching process of the surface could help in future investigations, as it has already been used to investigate the morphology of homogeneous thermoplastic-epoxy mixtures [22].

Due to the mediating effect of EDX, this method is restricted to a few microns. The extent of this averaging could be more accurately predicted by Monte Carlo simulations as used in [20]. On the basis of investigations on micrographs, it can be concluded that no interphase formation to the extent reported in [5,18] has taken place. Furthermore, the EDX analysis reveals that PA6 can be distinguished from epoxy resin due to its nitrogen content and PES mainly due to its sulfur content. Due to their similar composition compared to epoxy resin, PI and PEI are barely distinguishable from epoxy resin with EDX analysis. The chlorine occurring in the epoxy resin can be used to differentiate between film areas and epoxy resin areas but due to the high fluctuations and the very low chlorine content in the epoxy resin, more reliable identification of the interphase is not possible with the chlorine content either.

By comparing the experimental methodology with the literature, the deviating observations can be explained. Teuwen et al. worked with a mixture of the epoxy resins TGMAP and DGEBF with the hardener DDS and observed the dissolution necessary for interphase formation only above 120 ${}^{\circ }C$. Bruckbauer et al. also worked with elevated temperatures and used a heating ramp from 100 ${}^{\circ }C$ to 180 ${}^{\circ }C$. By measuring turbidity curves, Riccardi et al. [24] found demixing temperatures between about 40 ${}^{\circ }C$ and 50 ${}^{\circ }C$ for the DGEBA/DDS/MCDEA system with Ultem 1000 (PEI). They found out that PEI actually becomes chemically more compatible with the epoxy system DGEBA/MCDEA/DDS with increasing degree of curing, demixing only takes place due to an entropic influence of the increasing chain length [24]. The measurements of Bonneaud et al. [25] on the same resin system confirmed the findings of Riccardi et al. In addition, the authors found an upper critical demixing temperature for mixing with PES in contrast to mixing with PEI. PES thus demixed in the tests when a certain upper-temperature limit was exceeded. No lower temperature limit was reported in the tests, so that at least for the TGMAP/DGEBF/DDS system a solubility at room temperature can be assumed. While no interphase was to be expected for PA6 and PI, the absence of dissolution of the PEI film is probably due to the solution temperature, which was not exceeded in the tests. In the case of PES, it is possible that the dissolution at room temperature is simply not fast enough, so that the epoxy resin is cured before the film has dissolved to an observable extent. As the 3D-printed molds and the resin system used are not suitable for the fabrication of test specimens at curing temperatures above 100 ${}^{\circ }C$, the test specimens and the fabrication process have to be adapted for the investigation of the temperature influence.

The observations of the interface zone become particularly interesting when compared with the results of the mechanical experiments. Although PEI has not formed an extended interphase with the epoxy resin, the adhesion of both materials is so strong that the fracture of the test specimens takes place in the surrounding epoxy resin and not in the boundary zone. The stated strength is therefore only a lower estimate of the strength of the PEI-epoxy resin interface. The formation of a pronounced interphase, therefore, does not seem to be the necessary condition for high strength of the interface. At the same time, chemical compatibility (measured by solubility) does not seem to be a sufficient condition for high strength of the interface, since for PES a higher solubility in epoxy resin than PEI is reported by Vandi [20]. Since epoxy resins that cure at room temperature are also on the market and frequently used in resin infusion processes, the consideration of the adhesion after curing at room temperature is also relevant for sensor integration. Compared to PI, which is used as a standard flexible substrate for sensor integration [26,27], PEI offers a considerably improved adhesion. Sensors manufactured on PEI, therefore, promise less weakening of the fiber composite without the need for geometric measures such as the insertion of holes [13] or miniaturization. One question for further investigations would be whether the formation of a distinct interphase has a positive effect on the strength, i.e., whether a similarly strong bond can be achieved for PES as for PEI by partial dissolution of the film.

## 5. Conclusions

Various requirements have to be met before sensors can be integrated into load-bearing fiber composites. The sensors must withstand the harsh conditions during the manufacturing process, satisfy the electrical boundary conditions imposed by the surrounding composite and must not impair the mechanical properties of the composite in order not to weaken the main load-bearing function. Mechanical properties include resistance to crack propagation, often measured by the critical energy release rate, and delamination tendency, which can be measured by the mechanical strength of the interface. This article shows that one way to measure the strength of the contact surface is to examine specimens consisting of a film embedded in epoxy resin.

The strength of the interface is an important requirement for evaluating possible substrate materials for sensor production. The presented investigations show that, of the thermoplastic films considered, only PEI is suitable as a sensor substrate, since an increased tendency to delamination of the composites with integrated sensors can almost be excluded due to the strong bonding. Especially PES, which is promising from the point of view of solubility, showed the weakest bond to the room-curing epoxy resin in the tests, which leads to a decreased delamination resistance in a load-bearing fiber composite. An open question for future investigations is the influence of the temperature during the curing reaction on the dissolution of the PES film and therefore on the strength of the contact zone.

The microscopic examinations were initially meant to show the relation between interphase formation and high interfacial strength. However, no extensive interphase can be detected. A closer look at the used techniques reveals that they are not suitable for the investigation of interphases in the sub-micron range, in the case of EDX, light microscopy and fluorescence microscopy because of the resolution and in the case of electron microscopy because of the contrast. Light microscopy and fluorescence microscopy are particularly suitable for the detection of interphases in the two to three-digit micrometer range due to the high contrast. Although electron microscopy has low contrast, it could possibly exploit its resolution advantage better in combination with a suitable etching process. EDX analysis is completely unsuitable for the observation of concentration gradients on submicron scales due to its mediating effect. In addition, because of the similar atomic composition of PEI, PI and the epoxy resin used here, these materials cannot be distinguished by EDX analysis.

From the entirety of the investigations carried out here on the PEI-epoxy interface, it can be concluded that the formation of an extended interphase is not a necessary condition for high bond strength. On the other hand, high chemical compatibility is not a sufficient condition for high bond strength, as can be seen from the example of PES.

## Figures and Tables

**Figure 1 polymers-13-00330-f001:**
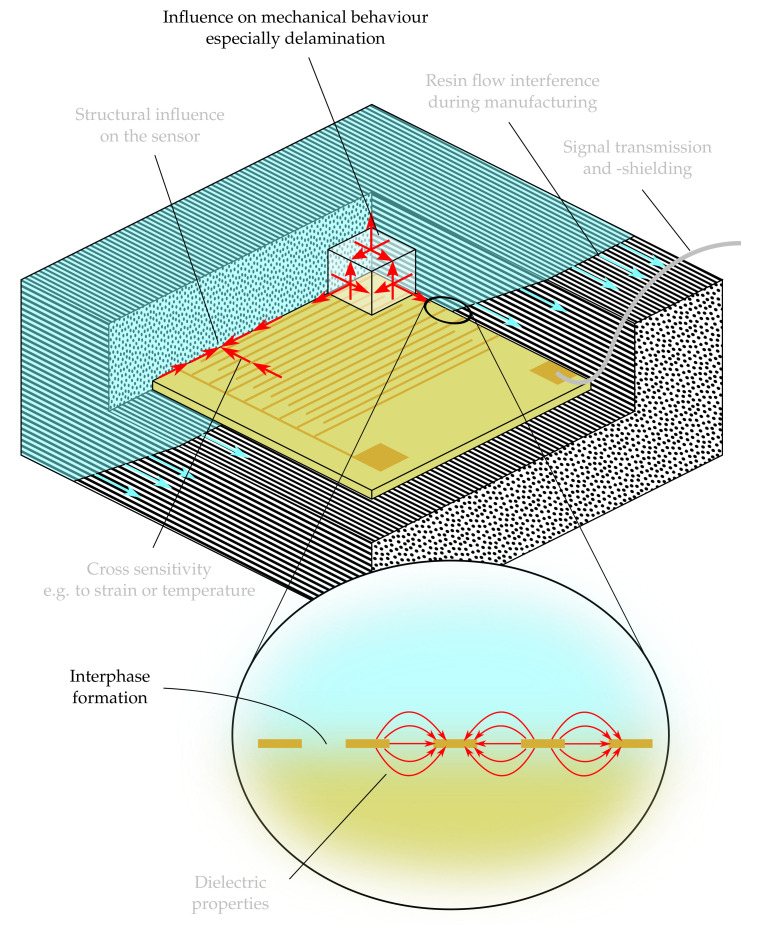
Questions arising from the integration of film sensors for cure monitoring purposes, modified from [12].

**Figure 2 polymers-13-00330-f002:**
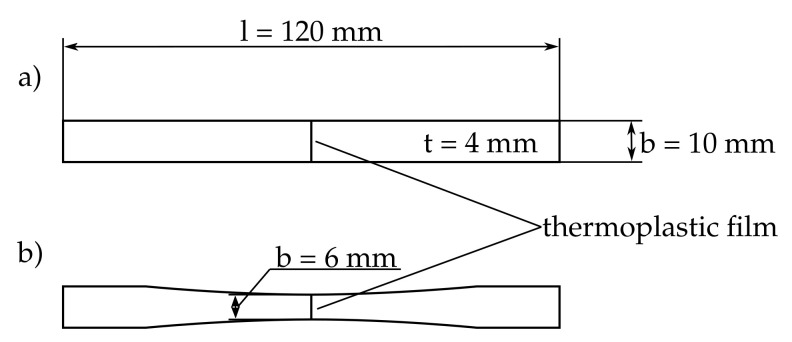
Geometry of the tensile specimens for interface investigation. (**a**) Tensile specimens which showed weak adhesion in preliminary experiments, (**b**) narrowed specimens for tensile tests on reference and polyetherimide (PEI).

**Figure 3 polymers-13-00330-f003:**
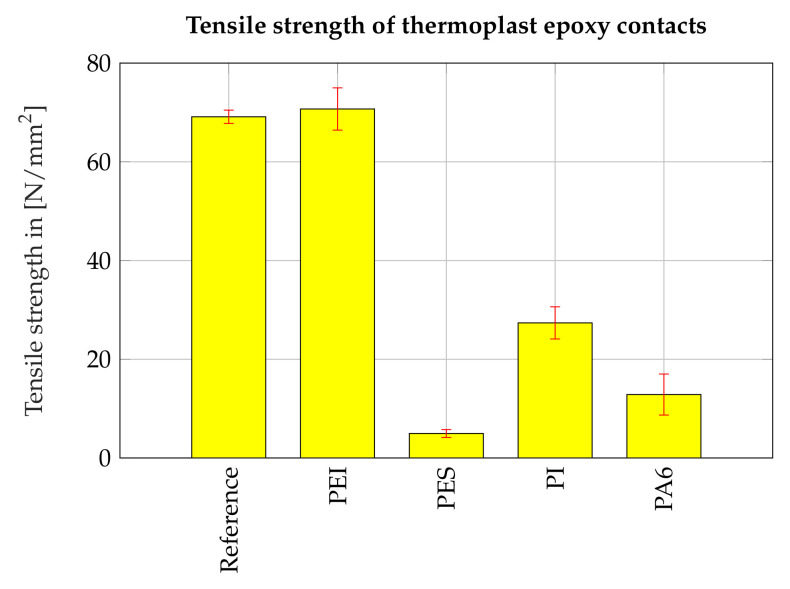
Tensile strength of the interface between epoxy resin and different thermoplastic films.

**Figure 4 polymers-13-00330-f004:**
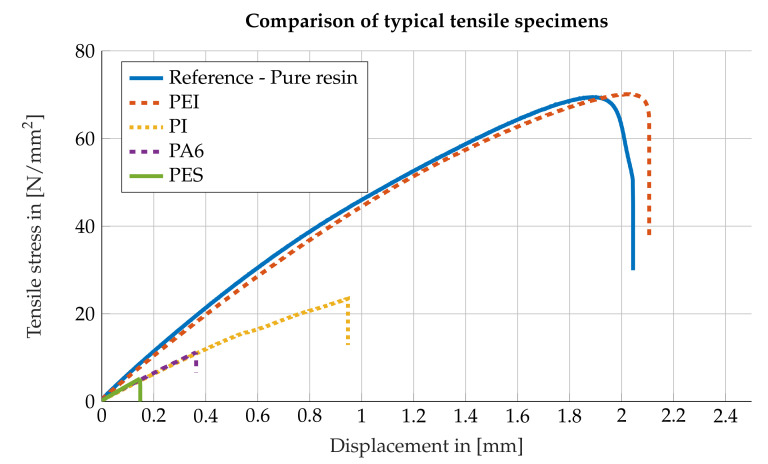
Comparison of typical curves of different types of specimen. It can be seen, that the PEI and pure resin specimens show the same pronounced arc before failure while the polyethersulfone (PES), polyamide 6 (PA6) and polyimide (PI) specimens show a sharp failure.

**Figure 5 polymers-13-00330-f005:**
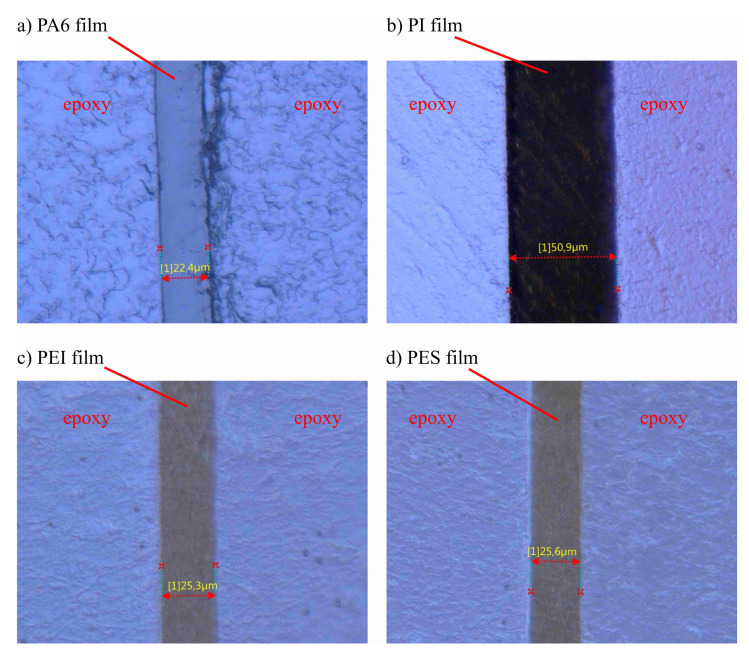
Light microscopic images of the interfaces between (**a**) PA6 and epoxy resin, (**b**) PI and epoxy resin, (**c**) PEI and epoxy resin and (**d**) PES and epoxy resin.

**Figure 6 polymers-13-00330-f006:**
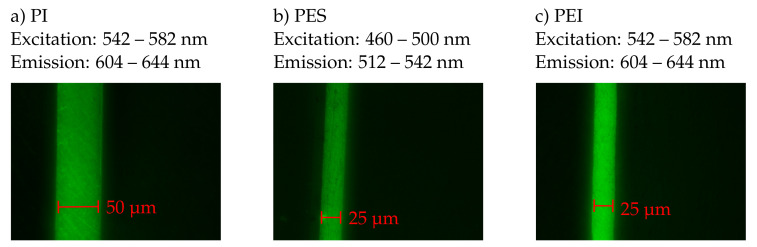
Fluorescence microscopic images of the interfaces between (**a**) PI and epoxy resin, (**b**) PES and epoxy resin and (**c**) PEI and epoxy resin.

**Figure 7 polymers-13-00330-f007:**
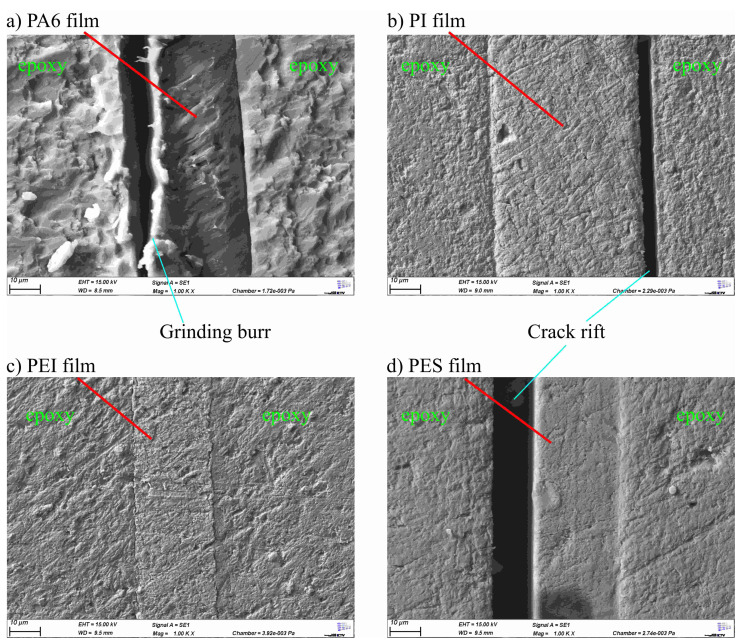
Scanning electron microscope images of the interfaces between (**a**) PA6 and epoxy resin, (**b**) PI and epoxy resin, (**c**) PEI and epoxy resin and (**d**) PES and epoxy resin.

**Figure 8 polymers-13-00330-f008:**
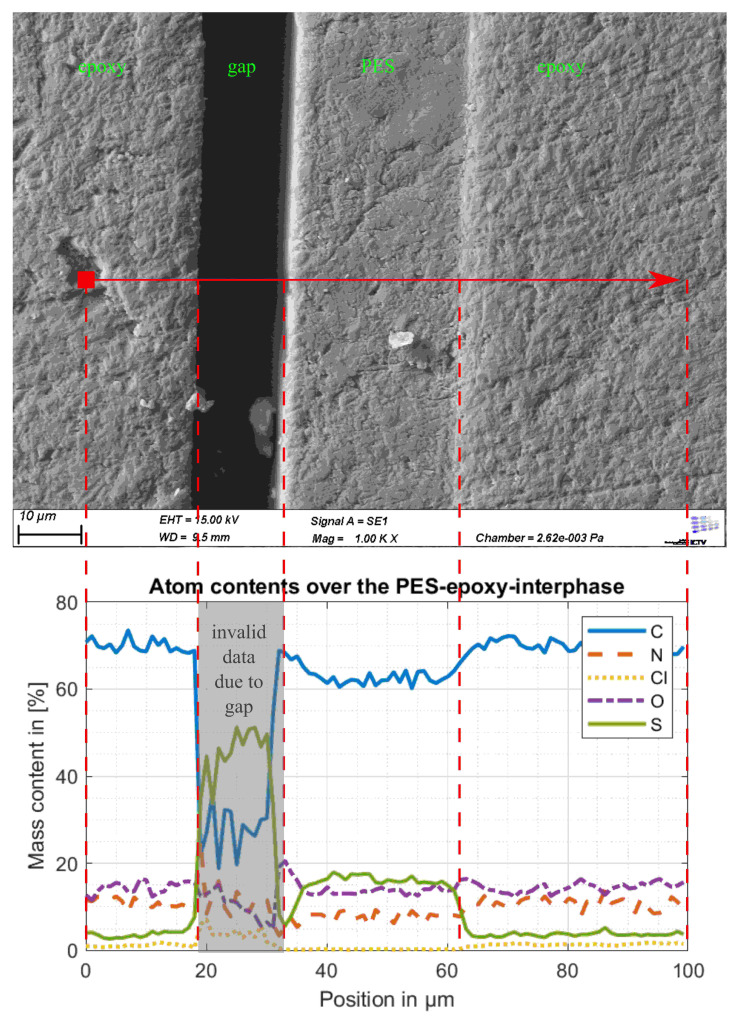
Scanning electron microscope (SEM) image of the interface between PES and epoxy resin compared with the energy-dispersive X-ray spectroscopy (EDX) results of a line scan along the red arrow.

**Figure 9 polymers-13-00330-f009:**
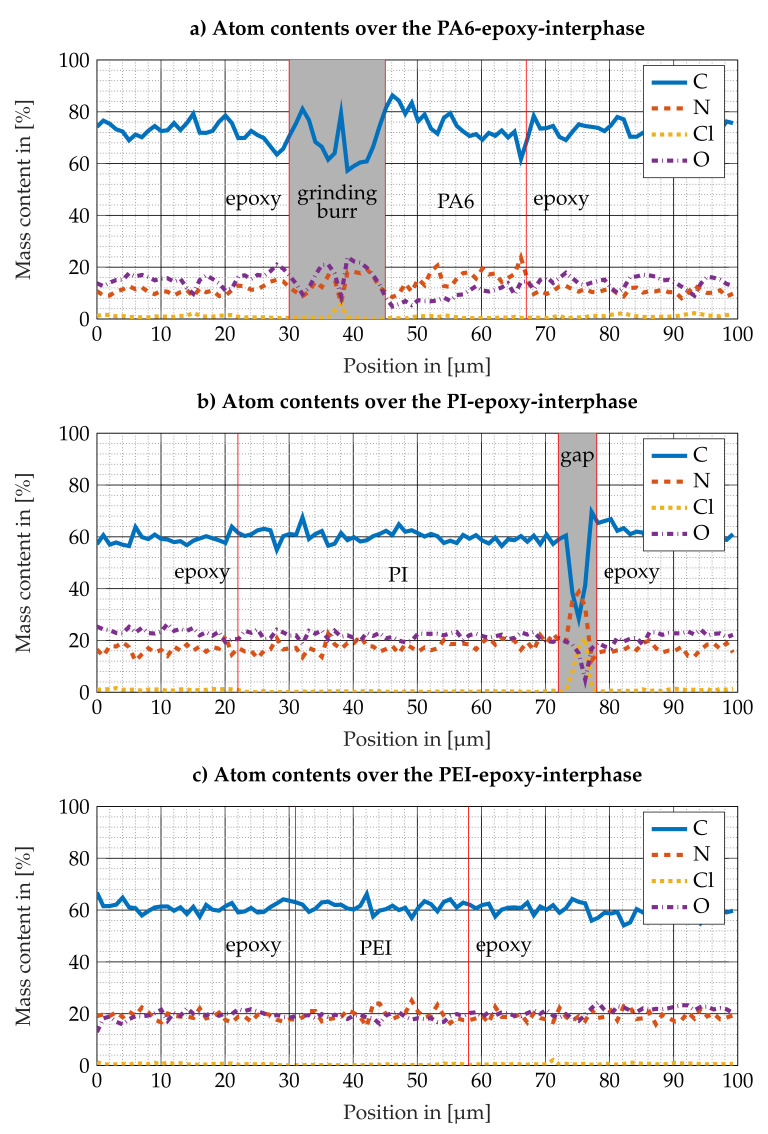
Comparison of EDX-Linescans along the interfaces between (**a**) PA6 and epoxy resin, (**b**) PI and epoxy resin and (**c**) PEI and epoxy resin. Invalid data due to a gap or grinding burr is marked with a grey background.

**Scheme 1 polymers-13-00330-sch001:**
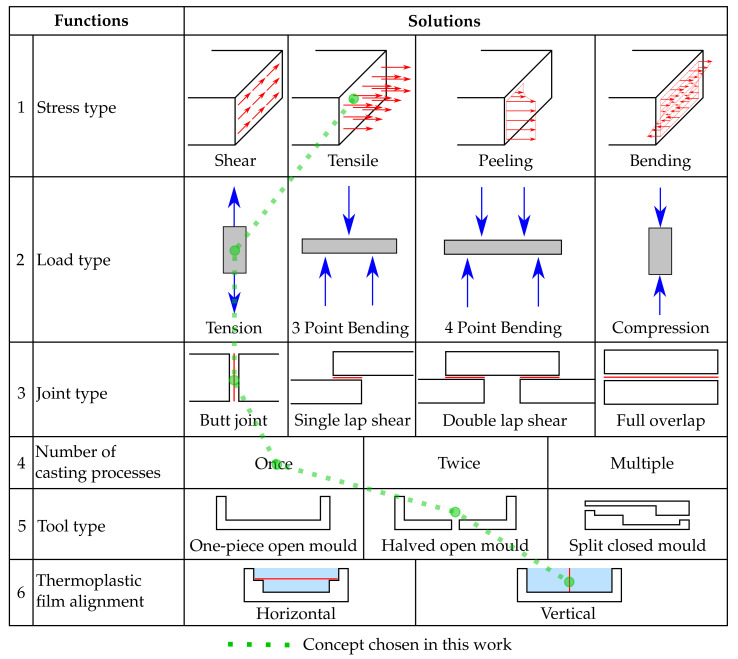
Morphological box used for specimen design and concept for specimens for interface strength investigation.

**Table 1 polymers-13-00330-t001:** Available filter configurations for fluorescence microscopy.

	405	YFP	L5	RH0	TXR	Y5
**Excitation [nm]**	375 to 435	490 to 510	460 to 500	541 to 551	542 to 582	590 to 650
**Emission [nm]**	450 to 490	520 to 550	512 to 542	565 to 605	604 to 644	662 to 738

## Data Availability

The raw data of the tensile experiments can be requested from the authors.

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
