# Peer review of "Tensile Strength and Structure of the Interface between a Room-Curing Epoxy Resin and Thermoplastic Films for the Purpose of Sensor Integration"

_polymers, 2021, doi:10.3390/polym13030330_

Round 1

Reviewer 1 Report

The work of kyriazis and coworkers has been submitted for consideration for publication in polymers.  The work relates to the strength of thin film/epoxy interfaces, and the authors spend some time discussing 'phase separation interface coarsening' in their introduction and discussion.  This is odd, because they see no interface broadening, and thus spend a lot of time discussing something they don't observe.  Their limited results seem trustworthy, particularly that they see no extended interfaces forming.  I see no reason why this should not be published here.

I do suggest several additions by the authors:

1.) given that you suppose that the difference in curing temperature explains why you see no interface, why not prove this by curing at higher temperatures?  This would make this a much more useful paper if you include evidence of a mechanism for your observation.

2.)figure 3 and figure 4 seem contradictory.  PI shows a higher tensile strength than PES and PA6 in figure 3, but figure four shows PI to be the lowest of all measurements.  Can you explain how this is, if all samples had the same dimensions?

3.) I recommend you calculate the energy release rate for these tests.  This would in principle be geometry independent, and thus a better 'number' to report in the literature.  This would also take into account the different thickness of the PI used by the authors, and allow a better comparison of the actual interfacial strength.

4.) while these tests are reasonable in the sense that they are an established method, I would suggest that they are not very relevant to actual conditions.  Films tend to be long, so failure is more likely to be by shear, or peel if the film is on a free surface.  The authors might consider testing in these additional geometries to make the work more useful to the 'thin electronics' community in particular.

Author Response

Dear Reviewer,

thank you very much for the positive and constructive feedback on my article.
I integrated the proposed improvements into the article.
A point-by-point answer is in the attachment.

Kind regards,
Alexander Kyriazis

Reviewer 2 Report

Comments to the Authors:

Manuscript title: Tensile strength and structure of the interface between a room-curing epoxy resin and thermoplastic films for the purpose of sensor integration

  1. Abstract: The results presented in the abstract are very general. Please make them more quantitative to realize better the amount of the influence of each film.
  2. Introduction: the main contribution of this study should be explained clearly in the last paragraph of the manuscript. The sentence “The present article contributes to close this gap” is not enough.
  3. Materials and methods: besides the name of the manufacturer of each material and device the country where it is made should be mentioned.
  4. Please show parameters t and b of Eq. 1 on the specimen shown in Fig. 2.
  5. Please add an explanation about the reference sample in the materials and method section. It is still not clear which sample was considered as a reference.
  6. Page 7: What do authors imply by “As the fractures occurred in the surrounding epoxy resin, the measured strength value only represents a lower estimate of the real strength of the interface.”? Does it mean that the fracture did not occur in the adhesive layer in all samples? If this the case so the results of the test that the fracture did not occur in the interface should be removed from the interface tensile strength results.
  7. 4: what is the meaning of the parameters on the horizontal axis in Fig. 4? I think it is better to be changed to traction or displacement. Moreover, which curve belongs to the reference sample?
  8. In my opinion, all samples should be made with the same geometry to have a meaningful comparison between the results. If the tapered sample works well with all materials, So, why all samples have not made with tapered geometry?

Author Response

(The authors gave the same response as above.)

Round 2

Reviewer 2 Report

All comments responded clearly and completely. The paper can be published in the journal.